# The Consortium for Genomic Diversity, Ancestry, and Health in Colombia (CÓDIGO): building local capacity in genomics and bioinformatics

Leonardo Mariño-Ramírez [1] ✉, Shivam Sharma[2,3], James Matthew Hamilton[2], Thanh Long Nguyen [2], Sonali Gupta[1,3], Aravinth Venkatesh Natarajan[2], Shashwat Deepali Nagar[2,3], Jay Landon Menuey[2], Wei-An Chen[2], Adalberto Sánchez-Gómez[4], José María Satizábal-Soto[4], Beatriz Martínez [5], Javier Marrugo[5], Miguel A. Medina-Rivas[6], Juan Esteban Gallo[1,3], I. King Jordan [2,3] & Augusto Valderrama-Aguirre [7] ✉

The Consortium for Genomic Diversity, Ancestry, and Health in Colombia (CÓDIGO) aims to build a community of Colombian researchers in support of local capacity in genomics, bioinformatics, and precision health. Here, we present the first CÓDIGO data release and the consortium web platform, including annotations for more than 95 million genetic variants from 1441 samples representing 14 populations from across the country. CÓDIGO samples show a wide range of African (16.7%), Indigenous American (32.8%), and European (50.6%) genetic ancestry components, with five distinct ancestry clusters. Thousands of ancestry-enriched variants, with divergent allele frequencies across clusters, show pharmacogenomic and clinical genetic associations. Examples include African ancestry-enriched variants associated with fast metabolism of the immunosuppressive drug tacrolimus and malaria resistance, and European ancestry-enriched variants associated with nicotine dependence and hereditary hemochromatosis. CÓDIGO reveals the nexus between ancestry and health in Colombia and underscores the utility of collaborative genome sequence analysis efforts.

The South American country of Colombia has a diverse, multiethnic population. The three largest ethnic groups in Colombia are Mestizo, Afro-Colombian, and Indigenous[1]. Previous studies of Colombian genetic ancestry have revealed three-way African, Indigenous American, and European admixture, with substantial regional heterogeneity in patterns of ancestry and admixture[2–8]. Colombia's ancestrally diverse population, together with robust academic, healthcare, and biotechnology sectors, makes it an ideal setting for the study of human population and clinical genomics. Moreover, Latino populations are vastly underrepresented in genomics research cohorts, and genomic studies of Colombian cohorts could help to close this research gap[9–11].

Genomic technologies are increasingly being applied to human populations and cohorts across Colombia, but these initiatives are dispersed with little effort to create a collaborative infrastructure. Neither best practices for genome analysis nor the results of genomic studies are widely shared among Colombian investigators. The Consortium for Genomic Diversity, Ancestry, and Health in Colombia (CÓDIGO) was created as a means to support local research capacity in genomics, bioinformatics, and precision medicine and to build a community among researchers working on the Colombian population and clinical genomics. The objectives of this study are to (1) provide an overview of CÓDIGO, (2) describe our analysis of the CÓDIGO data release 1.0, and (3) present the CÓDIGO database and web platform.

[1]National Institute on Minority Health and Health Disparities, National Institutes of Health, Bethesda, MA, USA. [2]School of Biological Sciences, Georgia Institute of Technology, Atlanta, GA, USA. [3]IHRC-Georgia Tech Applied Bioinformatics Laboratory, Atlanta, GA, USA. [4]Physiology Sciences Department, School of Health, Universidad del Valle, Cali, Colombia. [5]Molecular Genetics Lab., Institute for Immunological Research, University of Cartagena, Cartagena, Colombia. [6]Centro de Investigación en Biodiversidad y Hábitat, Universidad Tecnológica del Chocó, Quibdó, Chocó, Colombia. [7]Grupo Instituto de Investigaciones Biomédicas, Departamento de Ciencias Biological, Universidad de Los Andes, Bogotá DC, Colombia. ✉e-mail: l.marino.ramirez@gmail.com; a.valderramaa@uniandes.edu.co

CÓDIGO operates on a collaborative model whereby contributing investigators and laboratories share de-identified genomic variant data with the CÓDIGO development team. The primary genomic data are protected on a secure server, and no individual-level data or meta-data are released to the public. Summary statistics gleaned from analysis of the primary data are made publicly available via the CÓDIGO web platform. Contributing investigators collaborate on and receive credit for the web platform and any publications that arise from the CÓDIGO team's analysis of the data.

The CÓDIGO web platform aggregates data from Colombian genome projects, integrating genetic ancestry with population-specific variant allele frequencies and clinical associations, and shares summary statistics with the global scientific community. Our initial analysis of the CÓDIGO data is focused on the relationship between genetic ancestry and health outcomes. We consider how differences in genetic ancestry influence the diagnostic yield of Colombian genomic data, and we report the results of a genome-wide screen for ancestry-enriched variants that are associated with drug response and the risk of disease.

## Results
### CÓDIGO dataset

The CÓDIGO dataset release 1.0 is made up of 1441 Colombian genomic variant samples contributed by investigators from a variety of participating institutions across the country (Table 1 and Fig. 1). There are eight independent datasets included in the release, corresponding to fourteen distinct populations. CÓDIGO populations were sampled from diverse Afro-Colombian, Indigenous, and Mestizo ethnic groups. There are ten different Colombian indigenous communities represented in the CÓDIGO dataset. The majority of the samples ($n = 1122$) were characterized from the predominantly Mestizo population of Antioquia. The genomic dataset includes samples that were characterized by whole genome genotyping (WGG; $n = 793$), whole exome sequencing (WES; $n = 520$), or whole genome sequencing (WGS; $n = 128$). Together, these data sources contributed a total of 123,187,329 high quality genomic variants; merging and harmonization of these data yielded a final dataset of 95,254,482 non-redundant variants. This final variant set covers the union of all variants found among all the individual CÓDIGO datasets, each of which only contributes a fraction of the total number of CÓDIGO variants (Supplementary Fig. 3).

CÓDIGO genomic variant data were merged with variant data from global reference populations from Africa, the Americas, and Europe to infer patterns of genetic ancestry and admixture for CÓDIGO samples (Supplementary Table 1). Genomic principal component analysis (PCA) was used to characterize the genetic ancestry of the CÓDIGO cohort as a continuous variable. CÓDIGO samples fall in between African, Indigenous American, and European reference samples, consistent with ancestry contributions from all three sources and varying degrees of admixture from each (Fig. 2A).

The program ADMIXTURE was used to infer genetic ancestry fractions for the global reference samples and the CÓDIGO samples (Fig. 2B)[12]. CÓDIGO samples show a wide range of three-way African, Indigenous American, and European admixture. On average, CÓDIGO samples show 16.7% African, 32.8% Indigenous American, and 50.6% European genetic ancestry. K-means clustering was used to cluster CÓDIGO samples based on their three-way genetic ancestry fractions, and the elbow method yielded an optimal number of K = 5 ancestry clusters for the CÓDIGO samples (Fig. 2C and D; Supplementary Fig. 4). These include three clusters that are predominantly African-like ($n = 149$), Indigenous American-like ($n = 75$), and European-like ($n = 232$), as well as two admixed clusters (Admixed1, $n = 603$, and Admixed2, $n = 350$). One of the admixed clusters falls closer to the European-like cluster, and the other admixed cluster falls closer to the Indigenous American-like cluster with a minor African ancestry component. Almost all of the CÓDIGO samples are admixed to some degree.

### Ancestry and diagnostic yield

We evaluated the influence of genetic ancestry on the clinical utility of CÓDIGO genomic variant data by comparing the diagnostic yield of

**Table 1 | CÓDIGO genomic variant dataset**

| Source code[a] | Source description[b] | Department(s)[c] | PubMed[e] | Contributor[d] | n samples | n variants | Tech[f] | Array/depth[g] |
|---|---|---|---|---|---|---|---|---|
| CHG | Afro-Colombian from Chocó | Chocó | 27668076, 28855283 | Universidad Tecnológica del Chocó | 100 | 567,184 | WGG | Illumina HumanOmniExpress-24 |
| PLQ | Afro-Colombian from San Basilio de Palenque | Bolívar | 27725671 | Universidad de Cartagena | 34 | 9,779,781 | WGS | 37x |
| IND | Indigenous Arhuaco, Curripaco, Emberá, Guahibo, Inga, Kogi, Piapoco, Waunana, Wayuu communities | Arauca, Chocó, Guainía, Guajira, Magdalena, Meta, Putumayo, Vichada | 22801491, 31545791 | Universidad de Antioquia | 50 | 296,141 | WGG | Illumina 650Y |
| SIN | Indigenous Sinú community | Córdoba and Sucre | 31217584 | Universidad de Cartagena | 19 | 1,459,520 | WGG | Illumina Multi-Ethnic Genotyping Array |
| CLM | Mestizo Colombian from Medellín | Antioquia | 18369456, 26432245 | Universidad de Antioquia | 94 | 81,568,727 | WGS | 30x |
| MCM | Mestizo Colombian from Medellín | Antioquia | 30967898 | Universidad CES | 404 | 23,188,190 | WES | 100x |
| MCA | Mestizo Colombian from Antioquia | Antioquia | 31591465, 34650589 | Universidad CES | 624 | 526,935 | WGG | Illumina Global Screening Array |
| VDC | Mestizo Colombian from Valle del Cauca | Valle del Cauca | na | Universidad del Valle | 116 | 6,481,171 | WES | 30x |

Descriptive data is provided for the eight independent datasets, corresponding to fourteen populations, that make up release 1.0 of the CÓDIGO dataset.

[a]Three letter code for each source data set.
[b]Ethnicity and geographic origins for each source data set.
[c]Colombian administrative departments (i.e., states) for each source data set.
[d]Contributing institution for each source data set.
[e]PubMed identifiers for the manuscripts where each source data set is reported.
[f]Genomic characterization technology used for each source data set: whole genome genotyping (WGG), whole exome sequencing (WES), whole genome sequencing (WGS).
[g]Genotyping array used for WGG, or sequencing depth for WES and WGS.

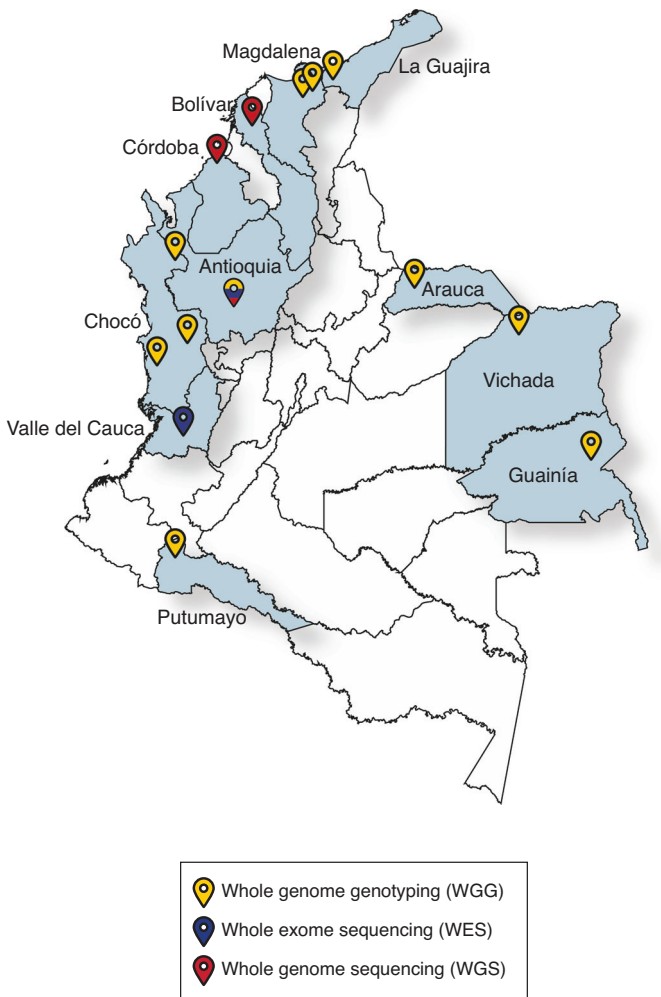

**Fig. 1 | CÓDIGO sample map.** Sample locations within Colombia are indicated with administrative departments shaded and labeled. The genomic technology used for each sample is color-coded as shown in the key: WGG (yellow), WES (blue), and WGS (red). The map file used to generate the image is freely distributed for MapSVG and is licensed under the Creative Commons Attribution 4.0 International Public License (URLs: https://mapsvg.com/maps/world and https://mapsvg.com/maps/colombia).

CÓDIGO genomic data across the five genetic ancestry groups characterized here. We quantified the diagnostic yield of CÓDIGO genomic samples as the percentage of variants with ClinVar annotations. ClinVar variant annotations were stratified into three pathogenicity classification groups: pathogenic/likely pathogenic, conflicting, and uncertain. Variants of uncertain pathogenicity were the most common, followed by variants with conflicting annotations, and pathogenic/likely pathogenic variants were the rarest as expected (Fig. 3). Overall, the Indigenous American-like (AME) ancestry group shows the lowest diagnostic yield and the African-like (AFR) ancestry group shows the second lowest. The European-like group shows an intermediate diagnostic yield, and the two admixed ancestry groups show the highest yield.

## Ancestry and variant associations

We performed a genome-wide screen for ancestry-enriched genomic variants in the CÓDIGO samples and compared the results of this screen to variant-health associations to explore the relationship between genetic ancestry and health in Colombia. We quantified ancestry-enriched variants as genetic variants that show anomalously large allele frequency differences between the five CÓDIGO sample ancestry clusters, and we considered variants with both pharmacogenomic and disease associations. The Fisher's

exact test used here is sensitive to changes in allele frequencies across ancestry clusters and shows thousands of variants with highly significant deviations from equal allele frequencies, as can be expected given the ancestral diversity of the CÓDIGO samples. We used a genome-wide significance threshold of $P < 5 \times 10^{-8}$ for the Fisher's exact test results and only considered variants with pharmacogenomic or disease associations that passed this threshold.

## Pharmacogenomic associations

We identified 585 significantly ancestry-enriched variants with pharmacogenomic associations across all evidence levels (Fig. 4A and Supplementary Data 1). There are 12 ancestry-enriched pharmacogenomic variants corresponding to evidence levels 1A, 1B, 2A, or 2B, which map to 10 different genes and are associated with responses to 9 different drugs (Table 2). The pharmacogenomic effect modes for these variants include associations with changes in drug metabolism and levels, with drug dependence, and with toxicity. We highlight pharmacogenomic associations of ancestry-enriched variants with tacrolimus metabolism, nicotine dependence, and methotrexate toxicity (Fig. 4B–D).

The ancestry-enriched variant found at chromosome 7 position 99,767,460 (rs4646437) maps to an intron of the Cytochrome P450 Family 3 Subfamily A Member 4 (CYP3A4) encoding gene. The alternate A allele for this variant is associated with rapid metabolism and decreased dose-adjusted trough concentrations of the immunosuppressive drug Tacrolimus compared to the reference G allele. The alternate A allele is positively associated with African ancestry and negatively associated with both Indigenous American and European ancestry (Fig. 4B). Consistent with these ancestry associations, the alternate A allele is found in higher frequency in the African-like (AFR) ancestry cluster compared to the Indigenous American-like (AME) and European-like (EUR) clusters. The admixed2 (ADX2) cluster also has a slightly higher frequency of the alternate A allele than the admixed1 (ADX1) cluster, consistent with its intermediate level of African ancestry.

The ancestry-enriched variant found at chromosome 15 position 78,590,583 (rs16969968) is a protein-coding missense variant of the Cholinergic Receptor Nicotinic Alpha 5 Subunit (CHRNA5) encoding gene. CHRNA5 is a ligand-gated ion channel that mediates rapid signal transduction at synapses when bound by nicotine or acetylcholine. The alternate A allele for this variant is associated with an increased risk for nicotine dependence compared to the reference G allele. The alternate A allele is positively associated with European ancestry and negatively associated with both African and Indigenous American ancestry (Fig. 4C). Consistent with these ancestry associations, the alternate G allele is found in higher frequency in the European-like (EUR) and Admixed (ADX1, ADX2) ancestry clusters compared to the African-like (AFR) and Indigenous American-like (AME) clusters.

The ancestry-enriched variant found at chromosome 1 position 11,796,321 (rs1801133) is a protein-coding missense variant of the Methylenetetrahydrofolate Reductase (MTHFR) encoding gene. MTHFR catalyzes a co-substrate for the methylation of homocysteine to methionine. The alternate A allele for this variant is associated with an increased risk of toxicity from the arthritis drug methotrexate compared to the reference G allele. The alternate A allele for this variant is positively associated with both Indigenous American and European ancestry and negatively associated with African ancestry (Fig. 4D). Consistent with these ancestry associations, the alternate A allele is found at a relatively low frequency in the African-like (AFR) ancestry cluster compared to all other clusters.

## Disease associations

We identified 24,837 significantly ancestry-enriched variants with clinical genetic annotations (Fig. 5A and Supplementary Data 2), 7 of which correspond to pathogenic or likely pathogenic ClinVar classifications (Table 3). We highlight disease associations of ancestry-enriched variants with malaria resistance, hemochromatosis type 1, and familial hypercholesterolemia (Fig. 5B–D).

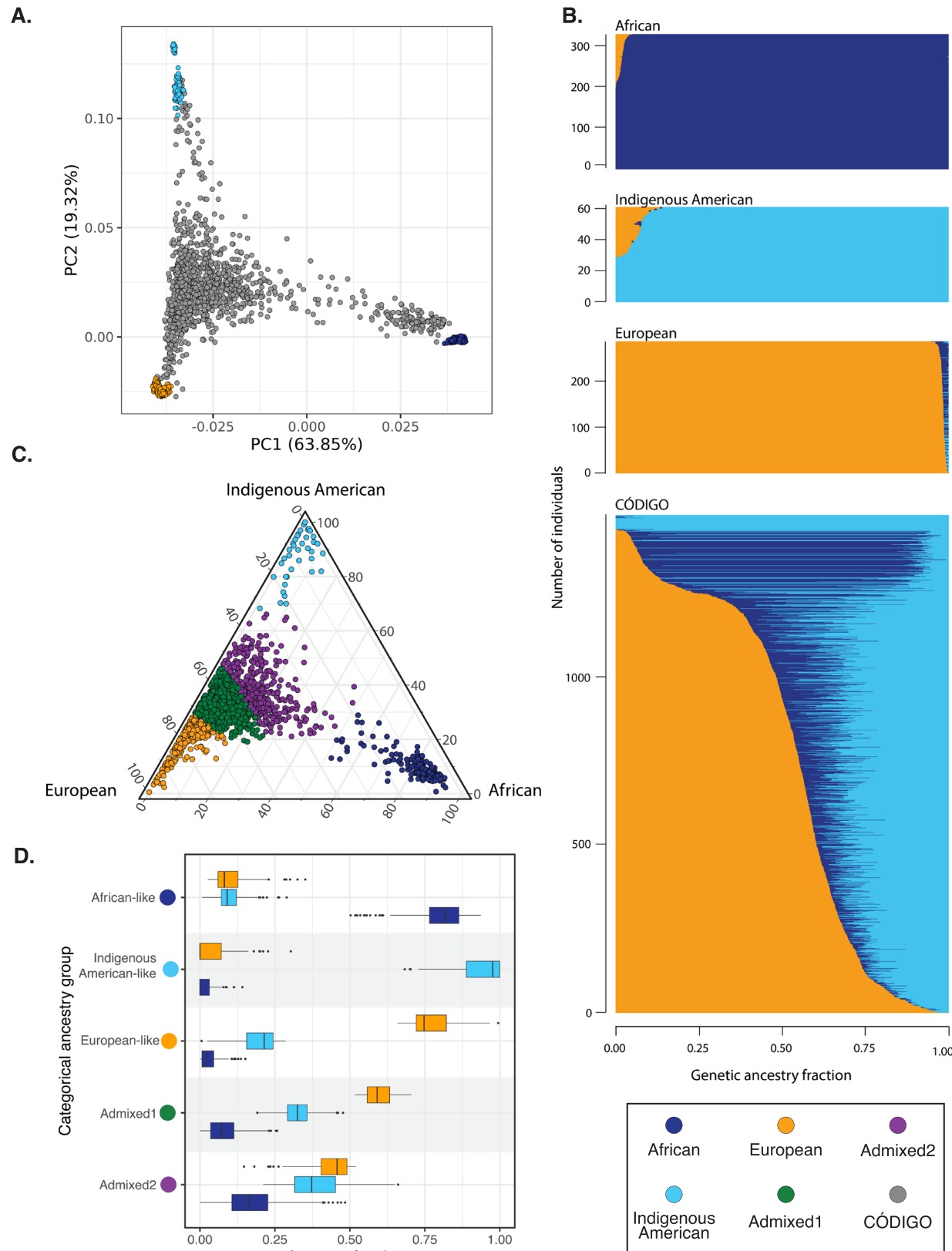

**Fig. 2 | Genetic ancestry and admixture. A** Principal component analysis of CÓDIGO sample genomic variant data. PC1 and PC2 are shown with the percent variation explained by each PC. Reference population samples are color-coded as shown in the key, and CÓDIGO samples are shown in gray. **B** ADMIXTURE plot with African, Indigenous American, and European ancestry fractions for each sample sown on the x-axis. **C** Ternary plot showing the relative proportions of African, Indigenous American, and European ancestry for each CÓDIGO sample. Samples are color-coded by their categorical ancestry group. **D** African, Indigenous American, and European ancestry fractions for each categorical ancestry group.

**Fig. 3 | Genetic ancestry and diagnostic yield.**
Percent of ClinVar annotated variants for categorical ancestry groups. ClinVar variant classifications: uncertain (blue), conflicting (orange), pathogenic or likely pathogenic (red). Ancestry groups: Admixed1 (ADX), Admixed2 (ADX2), African-like (AFR), Indigenous American-like (AME), and European-like (EUR).

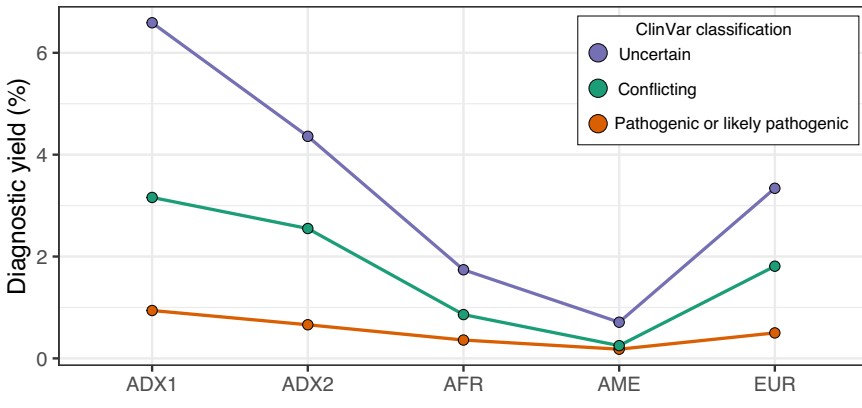

The ancestry-enriched variant found at chromosome 1 position 159,204,893 (rs2814778) maps to the 5' untranslated region (UTR) of the Atypical Chemokine Receptor 1, Duffy Blood Group (ACKR1) encoding gene. The alternate C allele for this variant is associated with resistance to malaria caused by *Plasmodium vivax* infection and reduced white blood cell count compared to the reference T allele. The alternate C allele is positively associated with African ancestry and negatively associated with both Indigenous American and European ancestry (Fig. 5B). Consistent with these ancestry associations, the alternate C allele is found in higher frequency in the African-like ancestry cluster (AFR) compared to the Indigenous American-like (AME) and European-like (EUR) clusters. The admixed2 (ADX2) cluster also has a slightly higher frequency of the alternate C allele than the admixed1 (ADX1) cluster, consistent with its intermediate level of African ancestry.

The ancestry-enriched variant found at chromosome 6 position 26,090,951 (rs1799945) is a protein-coding missense variant of the Homeostatic Iron Regulator (HFE) encoding gene, which regulates iron absorption. The alternate G allele is classified as pathogenic or likely pathogenic by the ClinVar database, compared to the reference C allele, and it is associated with the iron storage disorder hemochromatosis type 1. The alternate G allele is positively associated with European ancestry and negatively associated with African ancestry (Fig. 5C). No significant association was observed for Indigenous American ancestry. Consistent with these ancestry associations, the alternate G allele is found in the highest frequency in the European-like (EUR) ancestry cluster, followed by the admixed (ADX1 and ADX) clusters, with the lowest frequency seen for the African-like (AFR) cluster.

The ancestry-enriched variant found on chromosome 1 position 161,223,893 (rs5082) maps to the 5' promoter region upstream of the Apolipoprotein A2 (APOA2) encoding gene, a constituent of high-density lipoprotein particles. The alternate A allele is classified as pathogenic or likely pathogenic by the ClinVar database, compared to the reference G allele, and it is associated with increased plasma low-density lipoprotein (LDL) levels in familial hypercholesterolemia patients. The alternate allele is more common than the reference allele, and it only exerts its pathogenic effect when present in combination with an LDL receptor mutation[13]. The alternate A allele is positively associated with African ancestry and negatively associated with European ancestry (Fig. 5D). No significant association was observed for Indigenous American ancestry. Consistent with these ancestry associations, the alternate A allele is found in a higher frequency in the African-like ancestry cluster (AFR) compared to the European-like (EUR) and admixed (ADX1 and ADX2) clusters.

**Database and web platform**
The primary genomic variant data for CÓDIGO are stored securely and kept private as per our agreement with CÓDIGO data contributors. Secondary summary statistics calculated from the primary data, together with variant annotations, are made freely available via the CÓDIGO database and web platform at https://codigo.biosci.gatech.edu/. CÓDIGO summary statistics include variant allele counts and reference and alternate allele frequencies for each of the eight independent data sets. African, Indigenous American, and European genetic ancestry fractions are also provided for each dataset. Estimated reference and alternate allele frequencies are provided for each variant across all 32 Colombian administrative departments (states) and the capital district of Bogotá. CÓDIGO variant annotations include dbSNP identifiers, genomic location, and gene information, together with pharmacogenomic associations reported by the PharmGKB database and disease associations reported by ClinVar. Variant-specific hyperlinks for all external annotation sources used by the CÓDIGO database are also provided.

The CÓDIGO web platform front-end allows users to search the database by rsID, chromosome position, gene name, pharmacogenomic haplotype using the star allele nomenclature, or drug name (Fig. 6A). Searches can be performed using GRCh38 (hg38) or GRCh37 (hg19) human genome build coordinates. The CÓDIGO database and web platform are designed as variant-centric resources, and searching by rsID or chromosome position takes users to the final results webpage for each variant, including all of the summary statistics and variant annotations described in the previous paragraph. Searching by gene or pharmacogenomic haplotype takes users to a pre-results page that includes a list of all variants that map to the gene or haplotype with hyperlinks to each variant-specific final results page. Searching by drug takes users to a pre-results page that includes a list of all variants that are reported to be associated with that drug with hyperlinks to each variant-specific final results page. The underlying CÓDIGO database consists of tables with CÓDIGO variant information along with pharmacogenomic and clinical variant annotation tables, all of which are linked by variant integration tables (Fig. 6B).

## Discussion
In this report, we provide an overview of CÓDIGO – the Consortium for Genomic Diversity, Ancestry, and Health in Colombia. CÓDIGO is a community of researchers working on the population and clinical genomics of Colombian populations, and the goal of the consortium is to build local capacity in genomics, bioinformatics, and precision medicine. To help meet this goal, participating investigators contribute de-identified genomic variant data – characterized via WGS, WES, or WGG – which are harmonized and analyzed by the CÓDIGO development team (Supplementary Fig. 1). The results of this analysis, in the form of variant summary statistics, associations, and ancestry inferences, are publicly disseminated via the CÓDIGO web platform at https://codigo.biosci.gatech.edu/.

For release 1.0 of CÓDIGO, investigators from five participating institutions contributed 1,441 samples representing 14 distinct populations from 12 departments around the country (Table 1 and Fig. 1). Quality control, merging, and harmonization of these samples led to the characterization of variation patterns for more than 95 million genetic variants. Analysis of these variants underscores the ancestral diversity of the Colombian population along with the relevance of this diversity for pharmacogenomic and clinical genetic associations. There is a wide range of African, Indigenous American, and European admixture among CÓDIGO samples, and clustering on admixture patterns yields five ancestry groups (Fig. 2). Thousands of genetic variants show significant differences in allele

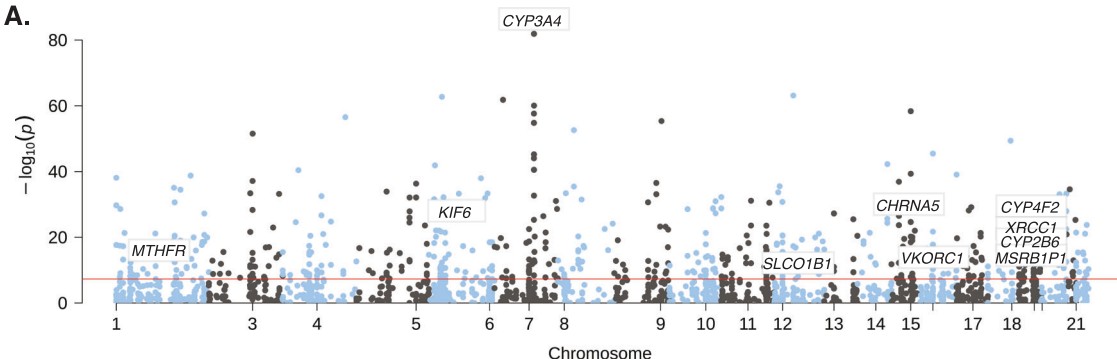

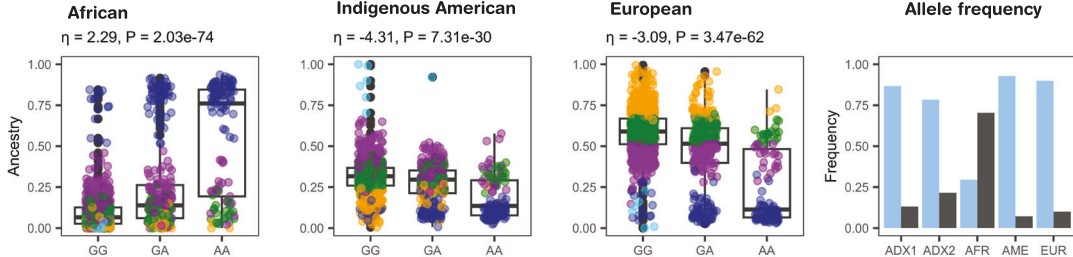

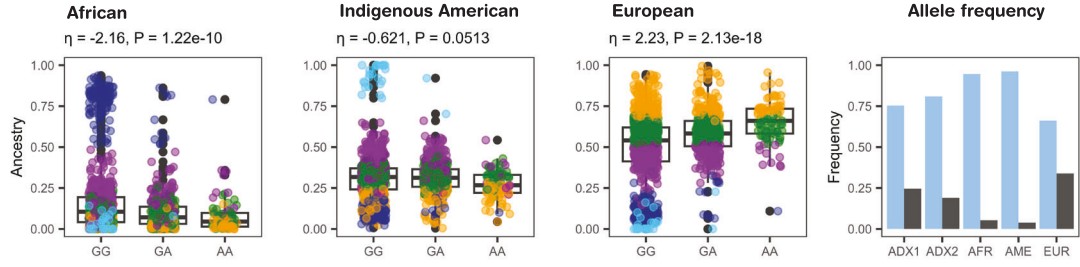

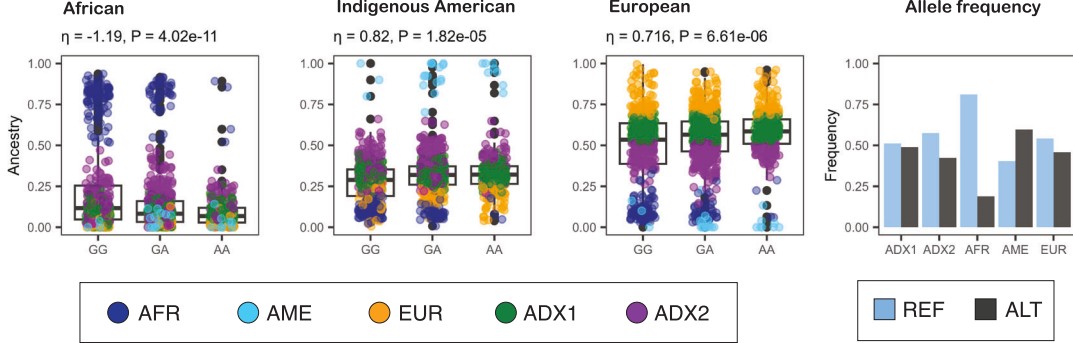

**Fig. 4 | Pharmacogenomic associations of ancestry-enriched variants.**
**A** Manhattan plot showing $-\log_{10}P$ values (y-axis) from Fisher's exact test of allele frequency differences across categorical ancestry groups, for PharmGKB annotated variants, with genome-wide significance threshold shown in red. **B–D** Examples of pharmacogenomic associations for ancestry-enriched variants. Regression between ancestry fractions – African, Indigenous American, and European– and genotypes are shown, with individuals color-coded by their ancestry groups (see Fig. 1). Reference (blue) and alternate (black) allele frequencies for ancestry groups.

frequencies between these ancestry clusters. 585 of these ancestry-enriched variants have pharmacogenomic associations and 24,837 have clinical genetic annotations. 12 of the pharmacogenomic associations for ancestry-enriched variants correspond to evidence levels 1 or 2 in the PharmGKB database (Table 2), and 7 of the clinical genetic annotations correspond to pathogenic or likely pathogenic variant classifications from the ClinVar database (Table 3). Analysis of CÓDIGO 1.0 data has recently been used to

explore the feasibility of CCR5 Δ32 stem cell transplant for HIV/AIDS treatment in Colombia[14].

The CÓDIGO results reported here contribute to a growing list of efforts to characterize patterns of genomic diversity, along with their phenotypic and clinical associations, for Latin American populations. For example, the Consortium for the Analysis of the Diversity and Evolution of Latin America (CANDELA) was one of the earliest efforts of this kind and

**Table 2 | Ancestry-enriched pharmacogenomic variant associations**

| Gene | Chr. | Position | rsID[a] | Pharmacogenomic Effect[b] | Evidence level[c] | Allele count[d] | Fisher's P-value[e] | $F_{ST}$[f] | Ancestry enrichment[g] |
|---|---|---|---|---|---|---|---|---|---|
| CYP3A4 | 7 | 99767460 | rs4646437 | Tacrolimus metabolism | 2A | 2406 | 1.28E−82 | 0.194 | AFR |
| KIF6 | 6 | 39357302 | rs20455 | Pravastatin metabolism | 2B | 1058 | 6.83E−33 | 0.139 | AFR |
| CHRNA5 | 15 | 78590583 | rs16969968 | Nicotine dependence | 2B | 2544 | 8.82E−21 | 0.035 | EUR |
| VKORC1 | 16 | 31093557 | rs9934438 | Warfarin metabolism | 1B | 2552 | 1.69E−18 | 0.032 | AME |
| MTHFR | 1 | 11796321 | rs1801133 | Methotrexate toxicity | 2A | 2574 | 2.18E−18 | 0.032 | AME |
| CYP4F2 | 19 | 15879621 | rs2108622 | Warfarin metabolism | 1A | 2628 | 1.34E−13 | 0.023 | EUR |
| VKORC1 | 16 | 31092475 | rs2359612 | Warfarin metabolism | 1B | 2428 | 1.66E−13 | 0.026 | AFR |
| XRCC1 | 19 | 43551574 | rs25487 | Platinum-based medication metabolism | 2B | 1326 | 1.60E−12 | 0.041 | AFR |
| VKORC1 | 16 | 31096368 | rs9923231 | Warfarin metabolism | 1A | 1918 | 4.69E−11 | 0.024 | EUR |
| CYP2B6 | 19 | 41012316 | rs28399499 | Antiretroviral metabolism | 2A | 2512 | 2.37E−10 | 0.028 | AFR |
| MSRB1P1 | 19 | 39252525 | rs8099917 | PEG-IFN alpha metabolism | 1B | 1762 | 3.14E−10 | 0.025 | ADX1 |
| SLCO1B1 | 12 | 21178615 | rs4149056 | Simvastatin toxicity | 1A | 2512 | 4.91E−08 | 0.013 | EUR |

[a]Single nucleotide polymorphism (SNP) identifier from the NCBI dbSNP database.
[b]Drug and effect mode for each pharmacogenomic association.
[c]PharmGKB database evidence level.
[d]Total number of alleles used for Fisher's exact test.
[e]P-value for 2 × 5 Fisher's exact test; 2 allele types (reference & alternate) X 5 ancestry clusters.
[f]Fixation index ($F_{ST}$) across the five ancestry clusters.
[g]Ancestry cluster that shows the highest alternate allele frequency.

includes more than seven thousand samples from Brazil, Chile, Colombia, Mexico, and Peru[4]. CANDELA has leveraged these data to discover cryptic ancestry components in Latin America and many novel genetic associations with physical appearance[15–20]. The EPIGEN-Brasil is one of the largest population genomics initiatives in Latin America, with genotyping and sequencing data for ~6500 participants from three Brazilian cohorts[21]. More recently, the Mexican Biobank project characterized WGG for more than six thousand participants sampled across all 32 states in Mexico[22], and the Mexico City Prospective Study characterized WGG and WES for ~140,000 participants and WGS for close to ten thousand participants[23]. On the commercial side, the company Galatea Bio is partnering with organizations across Latin America to collect genetic samples for the Biobank of the Americas[24,25]. They currently have about half a million samples and are aiming for 10 million by 2026. Despite these important efforts, genomic data from Latin American populations remain vastly underrepresented, and country-specific efforts like CÓDIGO are still needed to help build local capacity and support communities of genomics researchers.

The current version of CÓDIGO faces several limitations, which future efforts are aimed to address. The majority of the samples characterized for CÓDIGO release 1.0 are from the department of Antioquia and correspond to the Mestizo ethnic group, with primarily European genetic ancestry (Table 1 and Fig. 2). Subsequent releases of CÓDIGO will samples from more regions of Colombia, including recently launched collaborations focused on the Magdalena River Valley and the Valle del Cauca department. These regions are home to ethnically diverse communities and should allow for an increase in ancestral diversity of the CÓDIGO database. The current release does not include information on participant sex, and thus does not allow for sex-specific stratification and analysis of the cohort. We are planning to include sex information for a subset of participants in subsequent CÓDIGO releases.

The main analytical challenge for CÓDIGO was the harmonization of the disparate genomic variant data types that were contributed by participating institutions, particularly the WGG data, which were characterized using different array platforms and make up the majority of samples for the current version of CÓDIGO. The distinct variant sets for the arrays, compared to the sequencing variant data, resulted in a very small intersection of variants that are present in most or all of the datasets. Differences between variant sets captured by WGG arrays versus variants discovered by WES or WGS could also bias comparisons of diagnostic yield across ancestry groups.

As the size of the CÓDIGO dataset grows with future releases, more controlled comparisons of larger numbers of samples characterized using the same genomic technologies will be possible.

CÓDIGO was conceptualized and developed strictly as a genomics database, rather than a biomedical database (or a population biobank) that would include genomic data linked to participant demographic, anthropometric, socioenvironmental, and health outcome data. Thus, it is not currently possible to use CÓDIGO for precision medicine approaches, which require individual-level links between health outcomes, genetic data, and other potential risk factors. While CÓDIGO will remain primarily a genomic database, consistent with its main objectives, there may be efforts for individual contributors to include participant metadata of the kind described above to help power precision medicine studies on subsets of the database. One possible source of linked genomic, demographic, socioenvironmental, and health outcome data for CÓDIGO is genomic testing companies. Indeed, the Colombian healthcare system reimburses broadly defined genetic tests, and the majority of genome characterization efforts in the country, mostly clinical WES tests, are performed by companies rather than academic laboratories. We are actively recruiting corporate partners for subsequent releases of CÓDIGO with the aim of increasing our reach and scope. It should be stressed that any samples contributed by corporate partners will need to be properly consented for genetics research purposes.

CÓDIGO represents a nascent effort to nucleate currently dispersed Colombian genome sequence analysis projects in support of local capacity and community building. The CÓDIGO platform provides detailed information on millions of variants characterized from diverse populations around the country, relates ancestry to genetic determinants of drug response and disease, and underscores the potential of collaborative efforts to support genomic approaches to health and wellness in Latin America.

## Methods
### Genomic variant samples
CÓDIGO collaborating investigators and laboratories contributed de-identified genomic variant data generated by previous studies – characterized via whole genome genoytping (WGG), whole exome sequencing (WES), or whole genome sequencing (WGS) – with the CÓDIGO development team. De-identified genomic data were shared with the development team under the terms of data use agreements, and the primary data are

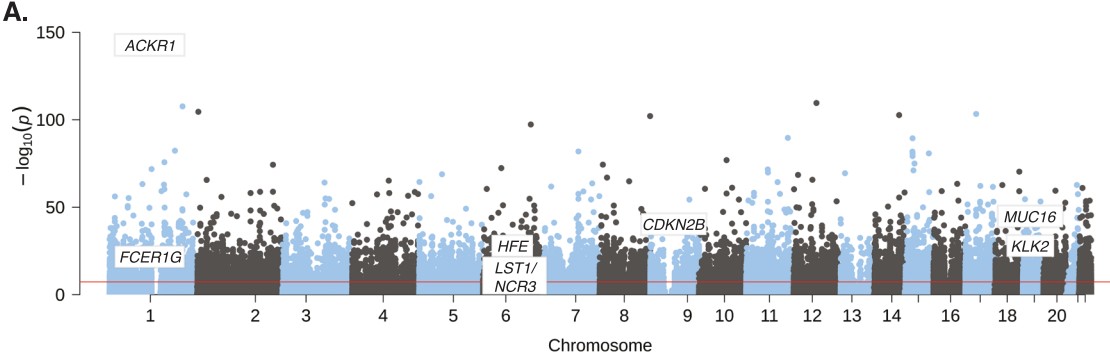

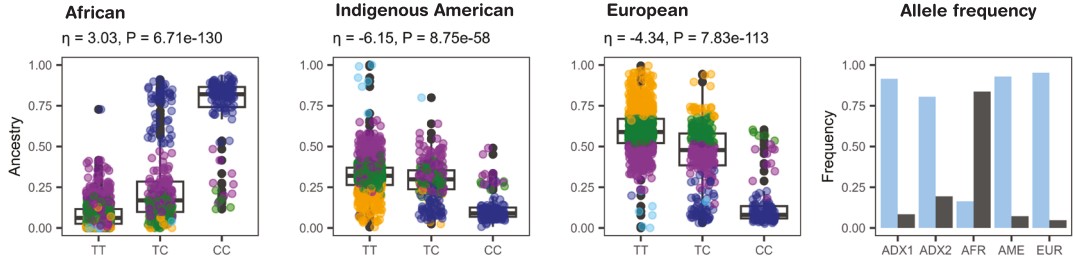

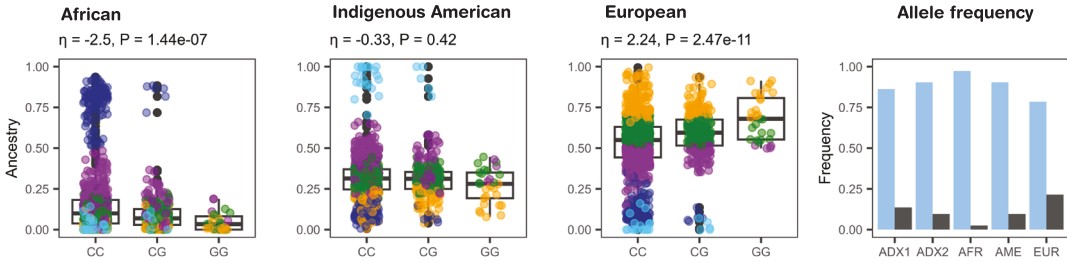

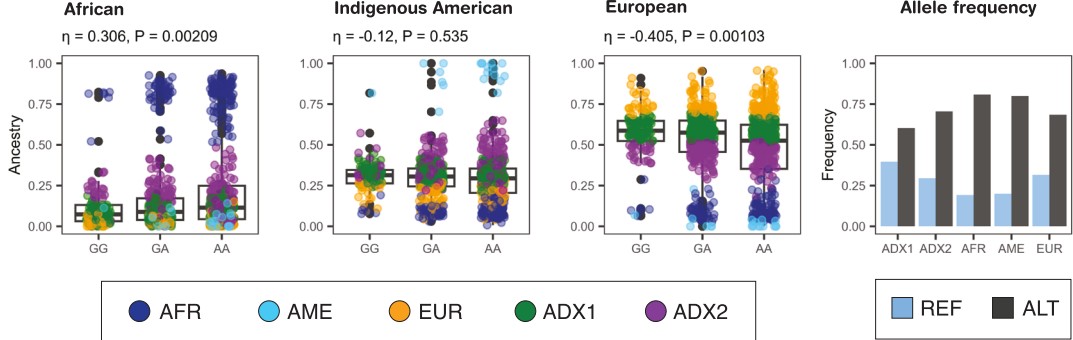

**Fig. 5 | Clinical genetic associations of ancestry-enriched variants. A** Manhattan plot showing $-\log_{10}P$ values (y-axis) from Fisher's exact test of allele frequency differences across categorical ancestry groups, for ClinVar annotated variants, with genome-wide significance threshold shown in red. **B–D** Examples of clinical genetic associations for ancestry-enriched variants. Regression between ancestry fractions – African, Indigenous American, and European – and genotypes are shown, with individuals color-coded by their ancestry groups (see Fig. 1). Reference (blue) and alternate (black) allele frequencies for ancestry groups.

protected in a secure server. Details on the methods used by CÓDIGO contributors to characterize genomic samples, study participant inclusion and exclusion criteria, and ethical review board approval have been previously published[2,7,26–35]. All previous study participants signed informed consent indicating their willingness to participate in genetic research. Details on the original studies where the samples were characterized and the ethical approvals for all CÓDIGO datasets can be found in the Supplementary Note on sample provenance and ethics approval. All previous studies that had

direct contact with participants conformed to the World Medical Association's Declaration of Helsinki ethical principles for medical research involving human subjects. Because the CÓDIGO genomic variant data were not collected specifically for this study, and no one on the study team has access to the subject identifiers linked to the primary data, this study is not considered human subjects research according to the NIH Revised Common Rule for the Protection of Human Subjects: https://grants.nih.gov/policy/humansubjects/hs-decision.htm.

**Table 3 | Ancestry-enriched disease variant associations**

| Gene | Chr. | Position | rsID[a] | Disease association[b] | ClinVar classification[c] | Allele count[d] | Fisher's P-value[e] | $F_{ST}$[f] | Ancestry enrichment[g] |
|------|------|----------|---------|------------------------|---------------------------|-----------------|---------------------|-----|------------------------|
| ACKR1 | 1 | 159204893 | rs2814778 | Malaria resistance; white blood cell count | Pathogenic; association; protective | 2570 | 4.31E−163 | 0.363 | AFR |
| CDKN2B | 9 | 22003368 | rs1063192 | Breast cancer | Likely pathogenic; protective | 1810 | 1.93E−20 | 0.044 | AFR |
| MUC16 | 19 | 8951121 | rs56971020 | Ovarian cancer | Likely pathogenic | 1042 | 3.73E−14 | 0.191 | AFR |
| HFE | 6 | 26090951 | rs1799945 | Hemochromatosis type 1 | Pathogenic | 2560 | 6.19E−13 | 0.023 | EUR |
| KLK2 | 19 | 50878521 | rs198977 | Acute myeloid leukemia | Pathogenic | 2592 | 1.02E−11 | 0.021 | AFR |
| FCER1G | 1 | 161223893 | rs5082 | Familial hypercholesterolemia | Pathogenic | 1934 | 8.58E−10 | 0.024 | AFR |
| LST1/NCR3 | 6 | 31593133 | rs2736191 | Malaria susceptibility | Pathogenic; risk factor | 762 | 6.02E−09 | 0.056 | AFR |

[a]Single nucleotide polymorphism (SNP) identifier from the NCBI dbSNP database.
[b]Disease or trait for which the variant is associated.
[c]ClinVar germline classification for the variant.
[d]Total number of alleles used for Fisher's exact test.
[e]P-value for 2 × 5 Fisher's exact test; 2 allele types (reference & alternate) X 5 ancestry clusters.
[f]Fixation index (FST) across the five ancestry clusters.
[g]Ancestry cluster that shows the highest alternate allele frequency.

## Variant merging and harmonization

Variant merging and harmonization were performed for 8 genomic variant datasets: 4 WGG, 2 WES, and 2 WGS (Supplementary Fig. 1). Custom variant file format conversion scripts and Plink version 1.9, with the fixed allele setting, were used to convert genomic variant datasets to Plink file formats bed/bim/fam[36,37]. Genomic variant identifiers were converted to Gnomad variant identifier format chr:pos:ref:alt, and Plink format files were converted to VCF format using Plink version 1.9. Genomic variant datasets from the human reference genome build GRCh37 (hg19) were lifted over to GRCh38 (hg38) using GATK version 4.0.10[38]. Lift over was performed using the recover swapped alleles flag to fix or remove variants with reference and alternate allele inconsistencies, and variants were compared to the NCBI dbSNP database to ensure reference and alternate allele consistency. Sample quality control was performed using kinship analysis with the program KING version 2.2.7 to remove duplicate samples and first-degree relatives[39], and samples with >99% variant missingness were removed using Plink version 1.9. Merging of dataset-specific VCF files was preformed using the merge function in bcftools version 1.17 with the -merge flag set to the ID column[40,41], resulting in a single union VCF file containing 95,254,482 variants for downstream analysis. The initial number of variants and the final number of variants retained after merging and harmonization for each dataset are shown in Supplementary Table 1.

## Genetic ancestry inference

CÓDIGO genomic variant data were compared to WGS-derived variant data for global reference samples from the 1000 Genomes Project (1KGP) and the Human Genome Diversity Project (HGDP)[28,42]. A total of 664 reference samples from African, Indigenous American, and European populations were used for genetic ancestry inference (Supplementary Table 2). CÓDIGO variant datasets were merged with reference sample variants using Plink version 1.9, keeping the allele order fixed. To select maximally overlapping variant sites among data sets, variants sites were ranked in decreasing order by the number of CÓDIGO data sets in which they are found, and the top 250,000 sites were retained. We previously found that 250,000 genome-wide sites were a sufficient number to perform robust principal component analysis (PCA) based ancestry inference on a diverse set of samples[43]. Principal component analysis (PCA) of the merged CÓDIGO and reference sample genomic variant data was performed using the smartPCA program as implemented in Eigensoft version 8.0.0[44,45]. PCA eigenvalues and eigenvectors were calculated for the reference sample variant data, and CÓDIGO samples were projected onto the reference sample PC-space.

Then, for the prioritized sites in each CÓDIGO variant dataset, biallelic variants common to the CÓDIGO and reference datasets were merged, with >75% missingness and <5% minor allele frequency variants removed from the merged datasets, followed by linkage disequilibrium (LD) pruning using Plink version 1.9. Individual CÓDIGO variant sets were processed separately to maximize the overlap between CÓDIGO samples and reference samples, and the same number of 250,000 sites was retained for each variant set to ensure comparability in ancestry inference across sets

The program ADMIXTURE version 1.3.0 was used to infer African, Indigenous American, and European ancestry fractions for CÓDIGO and global reference samples[12]. For the ADMIXTURE analysis, individual CÓDIGO datasets were processed separately to maximize the overlap between CÓDIGO samples and reference samples. Biallelic variants common to each CÓDIGO dataset and the reference datasets were merged, with >75% missingness and <5% minor allele frequency variants removed from the merged datasets, followed by linkage disequilibrium (LD) pruning using Plink version 1.9. ADMIXTURE was run on the merged and LD-pruned datasets in unsupervised mode for K = 3 ancestry components. CÓDIGO sample genetic ancestry fractions inferred by ADMIXTURE were analyzed using K-means clustering, with the Kmeans algorithm from the sci-kit learn Python library[46], to find the optimal number of categorical ancestry groups in the CÓDIGO dataset and to assign CÓDIGO samples to ancestry groups.

## Variant annotations and summary statistics

Harmonized genomic variant data were annotated using external database sources and used to generate variant summary statistics. Variants were annotated with information from the NCBI dbSNP, Gene, and ClinVar databases[47–49] and the PharmGKB database[50,51]. Variant positions, reference and alternate alleles, and gene locations were taken from Gnomad[52]. Variant positions are stored in both GRCh38 (hg38) and GRCh37 (hg19) human genome build coordinates. Variant allele frequencies were calculated for each data set using bcftools and bespoke scripts[40,41]. Estimated variant allele frequencies for Colombian departments were calculated using the ancestry-specific allele frequency estimation in admixed populations (AFA) method[53].

## Ancestry-health associations

ClinVar annotations were used to estimate the diagnostic yield of CÓDIGO genomic variant data across genetic ancestry groups. ClinVar annotations were stratified into three pathogenicity classification groups – uncertain, conflicting, and pathogenic/likely pathogenic – and the number of variants showing ClinVar associations was recorded for each ancestry group. For each of the $i = 1 - 5$ ancestry groups, the diagnostic yield (D) for each of

## A. Frontend design

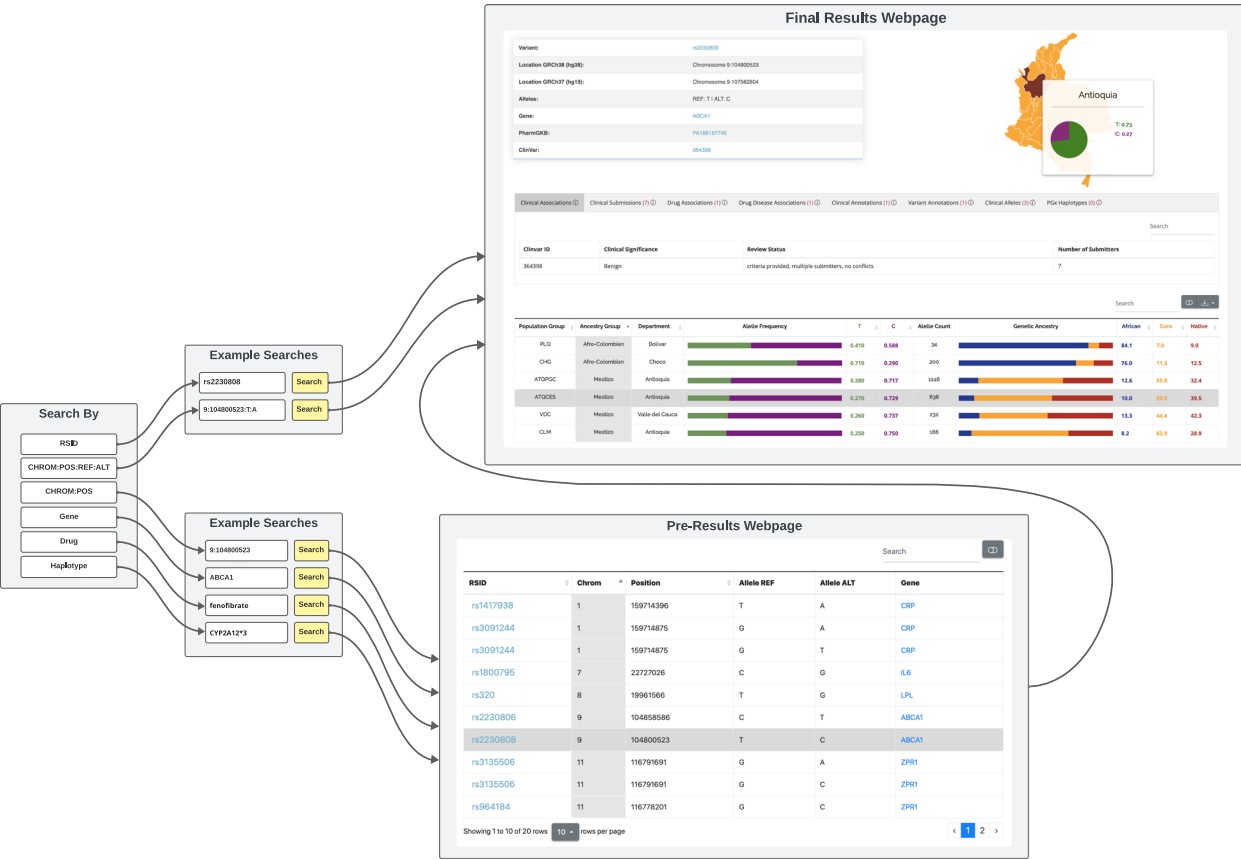

## B. Database schema

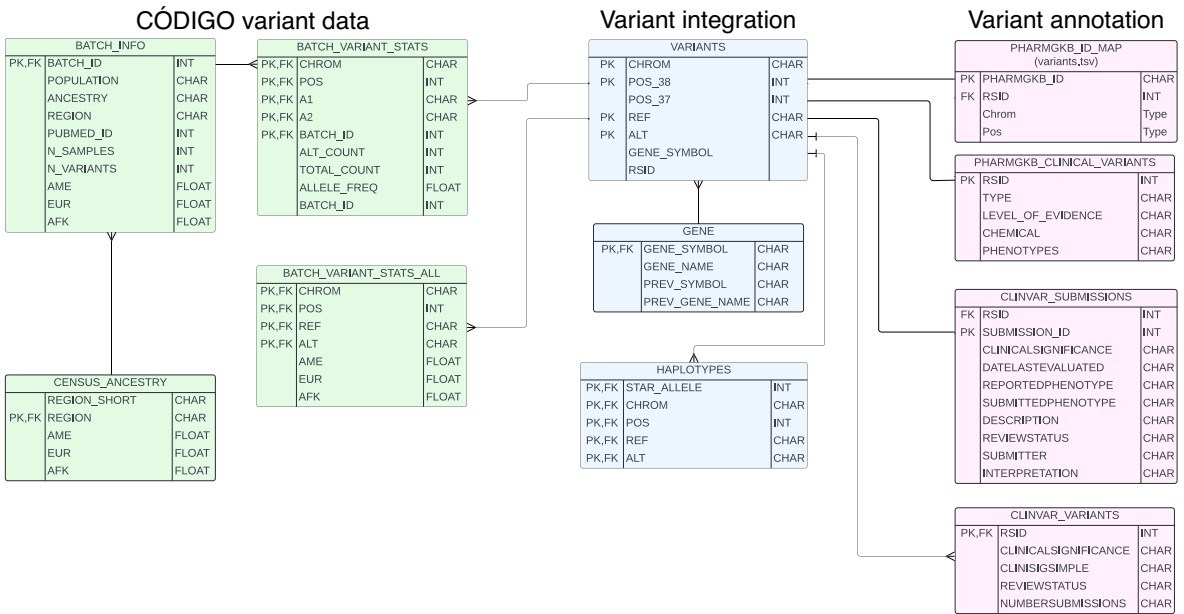

**Fig. 6 | CÓDIGO frontend design and database schema. A** User interface and user experience of the CÓDIGO webserver. **B** Data flow and storage for genomic variants and annotations.

the $p = 1 - 3$ pathogenicity classification groups was calculated as: $D_i^p = \#ClinVar\ variants/total\ \#\ variants$.

A 2 X 5 Fisher's exact test was used to identify ancestry-enriched variants as variants with anomalously high allele frequency differences between groups, with 2 rows for reference and alternate allele counts across 5 columns for each ancestry group: $\begin{bmatrix} ref_1 & ref_2 & \dots & ref_5 \\ alt_1 & alt_2 & & alt_5 \end{bmatrix}$. Variants that passed a genome-wide significance threshold of $P < 10^{-8}$ were screened for pharmacogenomic and disease associations using the PharmGKB[49,50] and ClinVar[49] databases.

Fixation index ($F_{ST}$) values for biallelic variants of interest were calculated as $F_{ST} = 1 - \bar{H}_S/H_T$[54]. $H_T$ is the total heterozygosity among all ancestry groups, calculated as: $H_T = 2pq$, where $p$ is the reference allele frequency among all groups and $q$ is the alternate allele frequency among all groups. $\bar{H}_S$ is the weighted average heterozygosity within the $i = 1 - 5$ ancestry groups, calculated as: $\bar{H}_S = \sum_{i=1}^{5} 2p_i q_i f(i)$, where $p_i$ and $q_i$ are ancestry group-specific reference and alternate allele frequencies and $f(i)$ is the proportion of individuals in each ancestry group.

Ancestry associations for biallelic variants of interest were calculated using Poisson regression, with model specifications as: $genotype \sim ancestry$, where $genotype \in [0, 1, 2]$ alternate alleles and $ancestry \in [0, 1]$. Poisson regression models were run separately for African, Indigenous American, and European ancestry fractions.

### Development stack
The CÓDIGO web platform was developed using Django, a high-level, open-source web framework written in Python following the Model-View-Template (MVT) architectural pattern (Supplementary Fig. 2)[55]. An SQL back-end database is used for data storage and user queries. The SQL database is stored locally with no API calls to external databases, facilitating faster maximum search speed, and user queries are processed by low-level calls to the database using custom parsers. The SQL scheme uses best practices, such as database normalization, ensuring that no genetic data is duplicated, and indexing of high-use querie,s ensuring rapid retrieval. The modular coding of the back-end is scalable and allows for ready addition and modification of front-end features requested by researchers and automated ingestion of new genetic and annotation data. Genetic variant annotations from external databases are automatically updated. The front-end web application was implemented using the Bootstrap framework[56].

### Statistics and reproducibility
The CÓDIGO web platform provides summary statistics – variant annotations, allele frequencies, and ancestry percentages – aggregated from Colombian genome projects. The results presented in Figs. 2–5 and Tables 2 and 3 represent additional analyses beyond the summary statistics provided on the web platform. Statistical analyses were used to associate ancestry-enriched genetic variants with disease and drug response phenotypes as detailed in the 'Ancestry-health associations' Methods subsection. Statistical analyses were conducted on the final merged and harmonized CÓDIGO dataset of 1409 samples and 95,254,482 variants. The association studies conducted do not include technical or biological replicates.

### Reporting summary
Further information on research design is available in the Nature Portfolio Reporting Summary linked to this article.

### Data availability
CÓDIGO summary statistics are made available on the CÓDIGO web platform: https://codigo.biosci.gatech.edu/ Access to the primary, de-identified genomic data are available on request from the individual data contributors. Requests should be addressed to CÓDIGO General Director Augusto Valderrama-Aguirre: a.valderramaa@uniandes.edu.co.

### Code availability
Code used to produce the published results is available on request from CÓDIGO General Director Augusto Valderrama-Aguirre: a.valderramaa@uniandes.edu.co.

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

## Acknowledgements

We gratefully acknowledge CÓDIGO participants for their contributions, without whom this research would not have been possible. L.M.R. and S.G. were supported by the Division of Intramural Research (DIR) of the National Institute on Minority Health and Health Disparities (NIMHD) at NIH (Award Number: 1ZIAMD000018). L.M.R. was supported by the National Institutes of Health (NIH) Distinguished Scholars Program (DSP). S.S., S.D.N., and I.K.J. were supported by the IHRC-Georgia Tech Applied Bioinformatics Laboratory (Award Number: RF383). A.V.A. was supported by a FAPA project (PVI0122029) from Universidad de Los Andes.

## Author contributions

L.M.R. and A.V.A. conceived of, led, and directed the project. L.M.R., J.E.G., I.K.J., and A.V.A. recruited participating investigators. I.K.J. and J.E.G. managed the CÓDIGO development team. A.S.G., J.M.S.S., B.M., J.M., M.A.M., and J.E.G. contributed genomic variants. S.S., J.M.H, T.L.N., S.G., A.V.N., S.D.N., J.L.M., W.A.C., and J.E.G. performed data analysis. J.M.H. developed the database. J.M.H., A.V.N., and W.A.C. developed the web platform. S.S., J.M.H., T.L.N., and J.E.G. produced figures and tables. I.K.J. wrote the manuscript.

## Competing interests

The authors declare no competing interests.
