## [Transparent Peer Review file · Communications Biology]

The Consortium for Genomic Diversity, Ancestry, and Health in Colombia (CÓDIGO): building local capacity in genomics and bioinformatics

Corresponding Author: Dr Leonardo Mariño-Ramírez

Version 0:

Reviewer comments:

Reviewer #1

(Remarks to the Author)

The study presented by Mariño-Ramírez et al. focuses on the new CÓDIGO web platform and aggregated data from eight genomic projects in Colombia. Therefore, the study seeks to address the lack of representation of Latin American populations in genomic research. The study also shows the potential of the datasets included in the consortium web platform to contribute to future research in population genetics, pharmacogenomics, and clinical genetic associations. However, the authors could improve the manuscript by addressing several comments.

Major comments:

Page 3: "Indigenous American (American hereafter) admixture"

The authors should refer to Indigenous American admixture/ancestry/etc. in the text, figures, and tables. The term "American ancestry" can be very confusing. The authors should consider the whole term "Native American" or "Indigenous American" ancestry.

Page 4: "Details on the methods used by CÓDIGO contributors to characterize genomic samples, study participant inclusion and exclusion criteria, and ethical review board approval have been previously published 2,7,12-18."

The authors could include a more detailed summary of the different datasets included in CÓDIGO, in particular, the cohorts or projects included and the ethical review board approvals. For instance, samples from the CLM-Colombian dataset included in the 1000 Genomes Project and Colombian individuals included in the PAGE Project were also included in CÓDIGO. However, that information is not clearly mentioned in the text or Table 1.

A map of Colombia highlighting the distribution of the samples, like the map included in the CÓDIGO web platform, will be extremely useful in this manuscript.

Table 1: The studied dataset does not represent the vast majority of Colombian territory, and one department is more overrepresented than the other departments/locations. The authors could mention in the text that 1,122 Mestizo Colombian individuals are from the Department of Antioquia, while the remaining 1,441 studied individuals are from the remaining departments included in this consortium.

Data availability: "CÓDIGO summary statistics are made available on the CÓDIGO web platform".

Apparently, access to de-identified genomic data will be impossible for the scientific community through CÓDIGO or current repositories, limiting future research and replicating the results presented in the manuscript.

Page 10: "We are actively recruiting corporate partners for subsequent releases of CÓDIGO with aim of increasing our reach and scope."

Apparently, the authors are contacting private genomic testing companies for future endeavours and research rather than sharing the data with the scientific community. Were the research participants properly informed about the participation of corporate partners in CÓDIGO?

Page 6: “merging and harmonization of these data yielded a final dataset of 95,254,482 non-redundant variants.”
In Table 1, the authors highlighted that they included WGG data (or genome-wide SNP data) for 774 individuals (half of the dataset), and the number of generated variants is 567,184. It’s difficult to understand that the merged and harmonised dataset has notably more variants than the maximum number included in the SNP-array dataset. Is this dataset after quality control?

Page 4: “Individual CÓDIGO variant datasets were merged with reference sample variants for 250,000 high quality sites using Plink”

The merged dataset between the CÓDIGO dataset and the WGS reference dataset is 250,000 SNPs. This seems to be a low number after merging with WGS data from the 1KGP and the HGDP. Is there a reason for that?

Page 6: “These include three clusters that are predominantly African-like, American-like, and European-like, as well as two admixed clusters”

Most of the CÓDIGO dataset consists of Mestizo Colombian individuals (n=1,122, Table 1). Therefore, the distinction between two admixed clusters with the same continental ancestries but more or less different proportions needs more discussion and justification in the text.

Minor comments:

The authors should elaborate more on their approach for “precision medicine”. This term is mentioned in the title, Introduction and Discussion section with one sentence (“the goal of the consortium is to build local capacity in genomics, bioinformatics, and precision medicine”). However, it’s not described in the rest of the manuscript. As the author mentioned: “CÓDIGO was conceptualized and developed strictly as a genomics database, rather than a biomedical database (or a population biobank) that would include genomic data linked to participant demographic, anthropometric, socioenvironmental, and health outcome data.”

Abstract: “Thousands of ancestry-enriched variants, with divergent allele frequencies across clusters, show pharmacogenomic and clinical genetic associations.”

Discussion: “Thus, it is not currently possible to use CÓDIGO for genetic association or epidemiological studies.”

Both statements seem to be contradictory.

Except for Figure 5A, the rest of the figures presented in the manuscript cannot be obtained/replicated using the CÓDIGO web platform.

Page 7: “We identified 585 significantly ancestry-enriched variants with pharmacogenomic associations across all evidence levels (Figure 3A).”

Here, the authors described results that were not included in the manuscript. The authors could include a supplementary table listing these variants and their pharmacogenomic associations.

The manuscript does not mention the number of individuals included in each admixed genetic cluster (ADX1 and ADX2).

Figure 1: The remarkably small standard errors for the ancestry fractions in Fig 1D don’t correspond with the large genetic variation observed in CÓDIGO individuals (Fig.s 1A-C).

Figures 3 and 4: The Manhattan plots did not include the names of the genes highlighted in the text and figures in the panel.

In the Discussion section, the authors mention other important consortiums for genomic studies in Latin America. However, the Brazilian EPIGEN-Brasil Initiative is missing here.

Is there a reason to include second-degree relatives in the analysed dataset?

Reviewer #2

(Remarks to the Author)

This manuscript by Mariño-Ramírez et al describe the creation of the Consortium for Genomic Diversity, Ancestry, and Health in Colombia (CÓDIGO), which aims to support local capacity building in genomics, bioinformatics, and precision medicine and facilitate collaborations among researchers working on Colombian population and clinical genomics. The manuscript describes the first CÓDIGO data release coming from 1,441 samples representing 14 populations from across Colombia and the web platform they have developed to facilitate data sharing and collaborations. Finally, they present results from their initial analysis of the CÓDIGO data focusing on the relationship between genetic ancestry and health outcomes. Their efforts help address existing disparities in the representation of Latin American populations in genomics research, by enabling local capacity building and supporting communities of genomics researchers within and outside of Colombia. The manuscript is well written, and the figures are clear. The sample size is adequate and a great starting point for their future efforts, and importantly as shown in Figure 1, it has a decent distribution of ancestral proportions and

admixture. The methods used to address admixture and to infer genetic ancestry are adequate. The statistical analysis used to examine associations are sound. The web platform is user friendly and will be an important resource for local and global investigators. Below I include a few minor comments/concerns that should be addressed by the authors.

1. Sequencing specifics are currently missing from Table 1. While readers can go search cited papers, it would be helpful to include an additional column with minimal information regarding the specific arrays, technology, and WGS depth used for each cohort. This may also help investigators make analysis decisions for future studies using the database.
2. Details are scarce on the level of overlap of WGG arrays used across studies, despite this heterogeneity in arrays being mentioned as a limitation. Again, including an Upset plot and/or supplementary table detailing this would be of interest to readers and future web users.
3. Likewise, additional information on the population cohorts, proportion of female vs male samples (and potentially enabling web users to perform analyses on F and M vs F or M may be useful down the line) for example, will be useful.

Version 1:

Reviewer comments:

Reviewer #1

(Remarks to the Author)

The manuscript presented by Mariño-Ramírez et al. focuses on the new CÓDIGO web platform and aggregated data from eight genomic projects in Colombia. The authors addressed all the questions from the reviews. However, some additional comments or clarifications could be taken into consideration to further improve this manuscript.

Regarding the Author's Response #13: The figures represent additional analyses beyond the summary statistics that are provided on the web platform. In this sense, the study and web platform can be considered to be complementary.

Since the focus of the manuscript is to present the CÓDIGO web platform, the authors should clearly indicate which figures can be achieved or found by the users using CÓDIGO and which ones were created by the authors using information not provided in CÓDIGO.

The term "precision medicine" seems to be misleading in the title. As the authors pointed out "it is not currently possible to use CÓDIGO for precision medicine approaches".

For this important dataset, the "Data availability statement" should provide detailed information for the contact person or the Data Access Committee of each "individual data contributor".

There are two Figure 5 in the manuscript. See: "Figure 5. Clinical genetic associations of ancestry-enriched variants." and "Figure 5. CÓDIGO frontend design and database schema."

Reviewer #2

(Remarks to the Author)

The authors have addressed all my concerns. Their platform will be a valuable contribution to the genetics community in Latin America.

Reviewers' comments:

Reviewer #1 (Remarks to the Author):

General Comment: The study presented by Mariño-Ramírez et al. focuses on the new CÓDIGO web platform and aggregated data from eight genomic projects in Colombia. Therefore, the study seeks to address the lack of representation of Latin American populations in genomic research. The study also shows the potential of the datasets included in the consortium web platform to contribute to future research in population genetics, pharmacogenomics, and clinical genetic associations. However, the authors could improve the manuscript by addressing several comments.

Response: We appreciate the reviewer's careful reading of our manuscript and the thoughtful suggestions for how it could be improved. We have addressed all of the comments and revised our manuscript accordingly as detailed in our point-by-point response below.

Major comments:

Comment #1.1: Page 3: "Indigenous American (American hereafter) admixture"

The authors should refer to Indigenous American admixture/ancestry/etc. in the text, figures, and tables. The term "American ancestry" can be very confusing. The authors should consider the whole term "Native American" or "Indigenous American" ancestry.

Response #1.1: We have changed "American" ancestry to "Indigenous American" ancestry throughout the text and figures as suggested.

Comment #3: Page 4: "Details on the methods used by CÓDIGO contributors to characterize genomic samples, study participant inclusion and exclusion criteria, and ethical review board approval have been previously published 2,7,12-18." The authors could include a more detailed summary of the different datasets included in CÓDIGO, in particular, the cohorts or projects included and the ethical review board approvals. For instance, samples from the CLM-Colombian dataset included in the 1000 Genomes Project and Colombian individuals included in the PAGE Project were also included in CÓDIGO. However, that information is not clearly mentioned in the text or Table 1.

Response #3: As suggested, we have provided additional details on the cohorts and projects included in the study and ethics approval for all CÓDIGO datasets in a new Supplementary Note on sample provenance and ethics approval. The original samples used for each CÓDIGO dataset and the sources of their ethics approval are described, and they are linked to PubMed identifiers for the studies where details on the ethics approval can be found (see page 4, paragraph 1, and the new Supplementary Note on sample provenance and ethics approval).

Comment #4: A map of Colombia highlighting the distribution of the samples, like the map included in the CÓDIGO web platform, will be extremely useful in this manuscript.

Response #4: As suggested, we have included a map highlighting the distribution of the samples as a new Figure 1.

Comment #5: Table 1: The studied dataset does not represent the vast majority of Colombian territory, and one department is more overrepresented than the other departments/locations. The authors could

mention in the text that 1,122 Mestizo Colombian individuals are from the Department of Antioquia, while the remaining 1,441 studied individuals are from the remaining departments included in this consortium.

Response #5: As suggested, we mention that the majority of the samples in CÓDIGO are Mestizo samples from Antioquia in the Results section (see page 6, paragraph 4). We also describe this a limitation of CÓDIGO release 1.0 and discuss how we plan to diversify the CÓDIGO datasets in subsequent releases in the Discussion section (see page 10, paragraph 4).

Comment #6: Data availability: “CÓDIGO summary statistics are made available on the CÓDIGO web platform”. Apparently, access to de-identified genomic data will be impossible for the scientific community through CÓDIGO or current repositories, limiting future research and replicating the results presented in the manuscript.

Response #6: The agreement with CÓDIGO contributors guarantees that all primary data will be securely held and only secondary, summary statistics will be released to the scientific community. We are bound by this agreement and need to honor it. Nevertheless, the reviewer makes a good point about how this may limit future research, and one of the main aims of CÓDIGO is to build a genomics research community in Colombia. Accordingly, contributors of CÓDIGO primary data have agreed in principle to provide access to primary, de-identified genomic data by request from individual contributors. We have added a statement describing this in the Data availability section (see page 12).

Comment #7: Page 10: “We are actively recruiting corporate partners for subsequent releases of CÓDIGO with aim of increasing our reach and scope.” Apparently, the authors are contacting private genomic testing companies for future endeavours and research rather than sharing the data with the scientific community. Were the research participants properly informed about the participation of corporate partners in CÓDIGO?

Response #7: This is a very good point. Discussions with genomic testing companies are ongoing, and we will need to ensure that any samples from that are used from testing companies are properly consented for genetics research. We have added a statement making this point to the Discussion section (see page 11, paragraph 2).

Comment #8: Page 6: “merging and harmonization of these data yielded a final dataset of 95,254,482 non-redundant variants.” In Table 1, the authors highlighted that they included WGG data (or genome-wide SNP data) for 774 individuals (half of the dataset), and the number of generated variants is 567,184. It’s difficult to understand that the merged and harmonised dataset has notably more variants than the maximum number included in the SNP-array dataset. Is this dataset after quality control?

Response #8: Yes, this is the final dataset after quality control. The 95,254,482 non-redundant variants represent the union of all sites across all CÓDIGO datasets. We have added text to the Results section to clarify this (see page 6, paragraph 4). The different CÓDIGO datasets overlap to different extents, so any given dataset will only make up a small fraction of the total number of variants in the final dataset. To clarify this, we have created an upset plot that shows the total number of variants and the extent to which variants sites overlap among datasets (see response to Reviewer 2 comment #2.2 below and new Supplementary Figure 3).

Comment #9: Page 4: “Individual CÓDIGO variant datasets were merged with reference sample variants for 250,000 high quality sites using Plink”. The merged dataset between the CÓDIGO dataset and the WGS reference dataset is 250,000 SNPs. This seems to be a low number after merging with WGS data from the 1KGP and the HGDP. Is there a reason for that?

Response #9: For the PCA-based ancestry analysis, we chose 250,000 sites to ensure that we were analyzing positions that showed the maximum amount of overlap between all CÓDIGO data sets and the reference samples. To select maximally overlapping variant sites among data sets, variant sites were ranked in decreasing order by the number of CÓDIGO data sets in which they are found and the top 250,000 sites were retained. We previously found that 250,000 genome-wide sites was a sufficient number to perform robust principal component analysis (PCA) based ancestry inference on a diverse set of samples. We have added a reference from our recent work to support this decision, and we have revised this section of the Methods to explain this better and to provide a rationale for the choice of 250,000 sites (see page 4, paragraph 3, and reference #29).

Comment #10: Page 6: “These include three clusters that are predominantly African-like, American-like, and European-like, as well as two admixed clusters”. Most of the CÓDIGO dataset consists of Mestizo Colombian individuals (n=1,122, Table 1). Therefore, the distinction between two admixed clusters with the same continental ancestries but more or less different proportions needs more discussion and justification in the text.

Response #10: We used K-means clustering of CÓDIGO sample genetic ancestry profiles, together with the elbow method, to identify the optimal number of ancestry clusters in the dataset. The elbow method yields K=5 as the optimal number of ancestry clusters, including the two admixed clusters. We elaborate on this in the Results section (see page 6-7, last and first paragraphs) and the elbow method is illustrated in Supplementary Figure 4.

Minor comments:

Comment #11: The authors should elaborate more on their approach for “precision medicine”. This term is mentioned in the title, Introduction and Discussion section with one sentence (“the goal of the consortium is to build local capacity in genomics, bioinformatics, and precision medicine”). However, it’s not described in the rest of the manuscript.

Response #11: We elaborate on our future plans for precision medicine in the Discussion section. As one of the study limitations, we mention that “... it is not currently possible to use CÓDIGO for precision medicine approaches, which require individual-level links between health outcomes, genetic data, and other potential risk factors.” and describe coming “... efforts for individual contributors to include participant metadata of the kind described above to help power precision medicine studies on subsets of the database.” (see page 11, paragraph 2).

Comment #12: As the author mentioned: “CÓDIGO was conceptualized and developed strictly as a genomics database, rather than a biomedical database (or a population biobank) that would include genomic data linked to participant demographic, anthropometric, socioenvironmental, and health outcome data.” Abstract: “Thousands of ancestry-enriched variants, with divergent allele frequencies across clusters, show pharmacogenomic and clinical genetic associations.” Discussion: “Thus, it is not currently possible to use CÓDIGO for genetic association or epidemiological studies.” Both statements seem to be contradictory.

Response #12: We agree that this was contradictory (thanks for catching that). We have corrected this and link it to future aims to include individual-level health outcome, and other risk factor data, in an effort to support precision medicine studies. The Discussion sentence has been modified to read “Thus, it is not currently possible to use CÓDIGO for precision medicine approaches, which require individual-level links between health outcomes, genetic data, and other potential risk factors.” (see page 11, paragraph 2)

Comment #13: Except for Figure 5A, the rest of the figures presented in the manuscript cannot be obtained/replicated using the CÓDIGO web platform.

Response #13: The figures represent additional analyses beyond the summary statistics that are provided on the web platform. In this sense, the study and web platform can be considered to be complementary.

Comment #13: Page 7: “We identified 585 significantly ancestry-enriched variants with pharmacogenomic associations across all evidence levels (Figure 3A).” Here, the authors described results that were not included in the manuscript. The authors could include a supplementary table listing these variants and their pharmacogenomic associations.

Response #13: As suggested, we provide two new supplementary tables that list the variants and their pharmacogenomic (revised Figure 4A) and clinical (revised Figure 5A) associations. (see Supplementary Tables 3 & 4).

Comment #14: The manuscript does not mention the number of individuals included in each admixed genetic cluster (ADX1 and ADX2).

Response #14: As suggested, we have added the numbers of individuals found in all genetic ancestry clusters to the Results section, including ADX1 and ADX2 (see page 7, paragraph 1).

Comment #15: Figure 1: The remarkably small standard errors for the ancestry fractions in Fig 1D don't correspond with the large genetic variation observed in CÓDIGO individuals (Fig.s 1A-C).

Response #15: We re-did this figure panel with box plots to better show the underlying ancestry distributions for each ancestry group, including their variation, following the editor's suggestion to follow the Communications Biology style guide (see revised Figure 2D).

Comment #16: Figures 3 and 4: The Manhattan plots did not include the names of the genes highlighted in the text and figures in the panel.

Response #16: We have added the names of the genes highlighted in the texts and figures to panel A of the revised figures 4 and 5.

Comment #17: In the Discussion section, the authors mention other important consortiums for genomic studies in Latin America. However, the Brazilian EPIGEN-Brasil Initiative is missing here.

Response #17: We added a description of the EPIGEN-Brasil cohort to the Discussion as suggested (see page 10, paragraph 3 and reference #52).

Comment #18: Is there a reason to include second-degree relatives in the analysed dataset?

Response #18: This decision was a trade-off between quality control and sample size. We removed first degree relatives (0.25 kinship coefficient) in our analysis to avoid inflating statistical noise in downstream analyses but included higher degree relatives to preserve sample sizes, especially for smaller endogamous populations like the Indigenous American groups we analyzed. Second-degree relatives show kinship coefficients of 0.0625 to 0.125, which is not expected to substantially bias the analyses we conducted.

Reviewer #2 (Remarks to the Author):

General Comment: This manuscript by Mariño-Ramírez et al describe the creation of the Consortium for Genomic Diversity, Ancestry, and Health in Colombia (CÓDIGO), which aims to support local capacity building in genomics, bioinformatics, and precision medicine and facilitate collaborations among researchers working on Colombian population and clinical genomics. The manuscript describes the first CÓDIGO data release coming from 1,441 samples representing 14 populations from across Colombia and the web platform they have developed to facilitate data sharing and collaborations. Finally, they present results from their initial analysis of the CÓDIGO data focusing on the relationship between genetic ancestry and health outcomes. Their efforts help address existing disparities in the representation of Latin American populations in genomics research, by enabling local capacity building and supporting communities of genomics researchers within and outside of Colombia. The manuscript is well written, and the figures are clear. The sample size is adequate and a great starting point for their future efforts, and importantly as shown in Figure 1, it has a decent distribution of ancestral proportions and admixture. The methods used to address admixture and to infer genetic ancestry are adequate. The statistical analysis used to examine associations are sound. The web platform is user friendly and will be an important resource for local and global investigators. Below I include a few minor comments/concerns that should be addressed by the authors.

General response: We appreciate the positive comments of the reviewer. We have addressed all of the reviewer's comments/concerns and revised our manuscript accordingly as detailed in our point-by-point responses below.

Comment #2.1: Sequencing specifics are currently missing from Table 1. While readers can go search cited papers, it would be helpful to include an additional column with minimal information regarding the specific arrays, technology, and WGS depth used for each cohort. This may also help investigators make analysis decisions for future studies using the database.

Response #2.1: As suggested, we added an additional column to Table 1 with information regarding the specific arrays and WES or WGS depth used for each cohort.

Comment #2.2: Details are scarce on the level of overlap of WGG arrays used across studies, despite this heterogeneity in arrays being mentioned as a limitation. Again, including an Upset plot and/or supplementary table detailing this would be of interest to readers and future web users.

Response #2.2: As suggested, we created an upset plot to show the extent of overlap among the CÓDIGO datasets. The main plot shows all samples, and the inset highlights the extent of overlap among the WGG samples (see new Supplementary Figure 3).

Comment #2.3: Likewise, additional information on the population cohorts, proportion of female vs male samples (and potentially enabling web users to perform analyses on F and M vs F or M may be useful down the line) for example, will be useful.

Response #2.3: Since we only had sex information for a subset of the cohort, the consortium decided not to include this information for CÓDIGO release 1.0. We have added a discussion of this as a limitation to the Discussion section of the manuscript and indicate our plans to include sex information for a subset of participants in subsequent CÓDIGO releases (see page 10, paragraph 4).

Reviewers' comments:

Reviewer #1 (Remarks to the Author):

General Comment: The manuscript presented by Mariño-Ramírez et al. focuses on the new CÓDIGO web platform and aggregated data from eight genomic projects in Colombia. The authors addressed all the questions from the reviews. However, some additional comments or clarifications could be taken into consideration to further improve this manuscript.

Response: We are pleased that the reviewer determined that we addressed all of the questions from the reviews. We appreciate the suggested additional comments and clarifications, all of which have been addressed as described in the responses to each point below.

Comment #1.1: Regarding the Author's Response #13: The figures represent additional analyses beyond the summary statistics that are provided on the web platform. In this sense, the study and web platform can be considered to be complementary.

Comment #1.2: Since the focus of the manuscript is to present the CÓDIGO web platform, the authors should clearly indicate which figures can be achieved or found by the users using CÓDIGO and which ones were created by the authors using information not provided in CÓDIGO.

Response #1.1 & 1.2: We agree that the study and web platform can be considered to be complementary. We have added a statement to the new Methods section on "Statistics and reproducibility" (page 11) to clarify this. "The CÓDIGO web platform provides summary statistics – variant annotations, allele frequencies, and ancestry percentages – aggregated from Colombian genome projects. The results presented in Figures 2-5 and Tables 2 and 3 represent additional analyses beyond the summary statistics provided on the web platform".

Comment #1.3: The term "precision medicine" seems to be misleading in the title. As the authors pointed out "it is not currently possible to use CÓDIGO for precision medicine approaches".

Response #1.3: As suggested, we changed the title to "The Consortium for Genomic Diversity, Ancestry, and Health in Colombia (CÓDIGO): building local capacity in genomics and bioinformatics".

Comment #1.4: For this important dataset, the "Data availability statement" should provide detailed information for the contact person or the Data Access Committee of each "individual data contributor".

Response #1.4: As suggested, we provided information for the contact person in the Data availability statement.

Comment #1.5: There are two Figure 5 in the manuscript. See: "Figure 5. Clinical genetic associations of ancestry-enriched variants." and "Figure 5. CÓDIGO frontend design and database schema."

Response #1.5: We corrected this to "Figure 6. CÓDIGO frontend design and database schema."

Reviewer #2 (Remarks to the Author):

General Comment: The authors have addressed all my concerns. Their platform will be a valuable contribution to the genetics community in Latin America.

Response: We are pleased that the reviewer indicated that we have addressed all of their concerns, and we appreciate the positive comment about our contribution to the genetics community in Latin America.